# The Effects of Lifestyle and Diet on Gut Microbiota Composition, Inflammation and Muscle Performance in Our Aging Society

**DOI:** 10.3390/nu13062045

**Published:** 2021-06-15

**Authors:** Barbara Strasser, Maike Wolters, Christopher Weyh, Karsten Krüger, Andrea Ticinesi

**Affiliations:** 1Medical Faculty, Sigmund Freud Private University, 1020 Vienna, Austria; 2Leibniz Institute for Prevention Research and Epidemiology–BIPS, 28359 Bremen, Germany; wolters@leibniz-bips.de; 3Department of Exercise Physiology and Sports Therapy, University of Giessen, 35394 Giessen, Germany; Christopher.Weyh@sport.uni-giessen.de (C.W.); karsten.krueger@sport.uni-giessen.de (K.K.); 4Geriatric-Rehabilitation Department, Azienda Ospedaliero-Universitaria di Parma, 43126 Parma, Italy; andrea.ticinesi@gmail.com; 5Department of Medicine and Surgery, University of Parma, 43126 Parma, Italy; 6Microbiome Research Hub, University of Parma, 43124 Parma, Italy

**Keywords:** aging, gut microbiota, immune system, muscle, nutrition, physical activity, protein

## Abstract

Living longer is associated with an increased risk of chronic diseases, including impairments of the musculoskeletal and immune system as well as metabolic disorders and certain cancers, each of which can negatively affect the relationship between host and microbiota up to the occurrence of dysbiosis. On the other hand, lifestyle factors, including regular physical exercise and a healthy diet, can affect skeletal muscle and immune aging positively at all ages. Accordingly, health benefits could partly depend on the effect of such interventions that influence the biodiversity and functionality of intestinal microbiota. In the present review, we first discuss the physiological effects of aging on the gut microbiota, immune system, and skeletal muscle. Secondly, we describe human epidemiological evidence about the associations between physical activity and fitness and the gut microbiota composition in older adults. The third part highlights the relevance and restorative mechanisms of immune protection through physical activity and specific exercise interventions during aging. Fourth, we present important research findings on the effects of exercise and protein as well as other nutrients on skeletal muscle performance in older adults. Finally, we provide nutritional recommendations to prevent malnutrition and support healthy active aging with a focus on gut microbiota. Key nutrition-related concerns include the need for adequate energy and protein intake for preventing low muscle mass and a higher demand for specific nutrients (e.g., dietary fiber, polyphenols and polyunsaturated fatty acids) that can modify the composition, diversity, and metabolic capacity of the gut microbiota, and may thus provide a practical means of enhancing gut and systemic immune function.

## 1. Introduction

Both the proportion of older people and the length of life continue to increase steadily in the Western societies. Nevertheless, living longer often comes with an increased risk of chronic diseases, including impairements of the musculoskeletal and immune system as well as metabolic disorders and certain cancers, each of which can negatively affect the relationship between host and microbiota. Although there is a dynamic equilibrium between the human host and the gut microbiota, age-dependent exposures can result in ecological disruptions of the gut microbiota up to the occurrence of dysbiosis. In fact, muscle wasting and poor nutritional status can facilitate inflammaging, which could cause a dysfunction of the immune system termed immunosenescence [1]. On the other hand, lifestyle factors, including physical activity and dietary habits, may affect skeletal muscle and immune aging positively [2]. This review will discuss the effects of physical activity and specific exercise and dietary interventions on gut microbiota composition, inflammation and muscle performance in the aging population. Usually only one of the mentioned lifestyle factors are addressed in intervention studies, while combined interventions are very rarely applied. Developing a broad understanding of the connection between microbiome, lifestyle and diet is a crucial factor for maintaining good health in our aging society.

## 2. Physiological Effects of Aging

### 2.1. The Aging Gut Microbiota

In healthy human beings, the composition and function of the intestinal microbiota is physiologically shaped by multiple environmental factors, among which geographical location, exercise, and dietary habits play a major role [3,4]. Host-related genetic factors also contribute to define its composition, but in a less effective way [5]. Gut microbial communities of healthy adults exhibit strong resilience to stressors, such as acute diseases, pharmacological treatments and changes in lifestyle or habits [6]. However, each stressor can leave a signature on the microbial composition in terms of biodiversity and balance between symbiont and pathobiont bacteria [7,8].

Aging, especially after the age of 70, is associated with distinctive changes in gut microbiota composition driven by the accumulation of the effects of multiple stressful events, physiological aging of the gastrointestinal system and altered lifestyle and dietary habits [8,9,10]. These changes generally include reduced biodiversity, increased representation of opportunistic Gram-negative bacteria and decreased representation of species with purported health-promoting functions [11]. These latter species may include Bifidobacteria, Lactobacilli and short-chain fatty acid (SCFA) producers, such as *Faecalibacterium prausnitzii, Eubacterium* spp., *Roseburia* spp., and *Ruminococcus* spp. [12].

Inter-individual differences in gut microbiota composition also tend to be emphasized in the older age, as suggested by a recent systematic review and meta-analysis of 27 human studies [13]. Resilience to stressors remains an important characteristic of the gut microbiota even in the late life. However, the pace and magnitude of recovery of the previous microbiota composition after an insult, such as a course of oral antibiotic treatment, depend on the environment of living and general health status [14]. 

In fact, in the relationship between aging and gut microbiota, age should be considered as a biological, and not chronological, entity. Centenarians are generally regarded as a biological model of successful aging, for their capacity of reaching the extreme decades of life in relatively good health [15,16]. Several studies have shown that the gut microbiota composition of centenarians of different ethnicity and living in different geographical locations exhibits elevated biodiversity and representation of bacteria, including *Firmicutes*, *Bifidobacteria*, and SCFA producers, with purported anti-inflammatory and homeostatic properties compared to that of younger elderly or younger adults [17,18,19,20,21]. These differences in composition are reflected in a different functionality of the microbiota, as determined by shotgun metagenomics sequencing and metabolomics on fecal samples [22]. Interestingly, these characteristics are similar to those observed in subjects in the 65–80 age range, supporting the concept that gut microbiota composition reflects biological rather than chronological aging [23].

Conversely, the gut microbiota composition of older persons with frailty or mobility-limitations exhibits various degrees of dysbiosis, with reduced species richness and imbalance between opportunistic pathogens and taxa with anti-inflammatory properties [24,25,26]. Studies conducted in nursing homes have shown similar characteristics of the composition of the residents’ microbiota, partly attributable to fixed dietary regimen, limited mobility and reduced contacts with the external environment [27,28]. The most severe forms of dysbiosis have been documented in older multimorbid patients admitted to hospital, where acute illness, underlying chronic diseases, polypharmacy and forced inactivity play as powerful disruptors of the intestinal microbial ecology [29,30]. Interestingly, similar changes of the microbiota composition have been observed also in centenarians with health status decline and approaching the end of life [31].

These findings support the hypothesis that gut microbiota composition reflects the aging trajectory. However, it is still unclear to what extent the microbiota represents an active player in the aging process or, instead, simply a biomarker of aging [8]. Fecal microbiota transplantation studies conducted on animal models have shown promising results, supporting the possibility of influencing the aging process through modulation of the microbiota [32,33]. However, the administration of probiotics and/or prebiotics in older human beings has rarely produced clinically significant improvements, suggesting that lifestyle intervention may be the best way to modulate the microbiota and host–microbe interactions [34].

The intestinal microbiota is however involved in the onset of several age-related chronic diseases and syndromes [35]. For example, it can modulate several pathophysiological processes involved in the onset and progression of neurodegenerative diseases, including dementia and Parkinson’s disease, through the so-called “gut–brain axis” [36,37]. It can also be involved in the complex mechanisms leading to age-related muscle wasting and sarcopenia [38,39], as recent evidence from small groups of patients suggests [40,41].

At the current state of knowledge, the gut microbiota should thus be considered at the cross-road between environmental influences and the host health. It contributes to the pathophysiological mechanisms leading to chronic illness and geriatric giants, such as frailty, sarcopenia, and cognitive impairment. At the same time, the microbiota is in-fluenced by lifestyle changes that are associated with the onset of such conditions. Therefore, the microbiome composition and function could reflect the aging trajectory, with maintenance of biodiversity and balance between symbionts and pathobionts in those who age in good health, and dysbiosis in those who age with frailty or disability (Figure 1).

### 2.2. The Aging Immune System

Like all other organs, the immune system is also affected by an aging process. Clinically, age-associated changes occurring in the immune system are manifested in an increased susceptibility to infections and prevalence of chronic diseases. For example, the mortality rate from ambulatory-acquired pneumonia increases slightly from the age of 35. At the cellular level, progressive immunaging is reflected by a reduction of numerous leukocyte functions as well as by altered composition of the leukocyte subpopulations. Here, a shift towards more pro-inflammatory cell types in the case of innate immune defense or towards more differentiated cell types in the case of adaptive immunity is evident [42]. In addition, there is a slight increase in pro-inflammatory cytokines in the blood, referred to as “inflammaging”. The immunological changes in healthy aging humans are also interpreted as physiological and age-appropriate remodeling adapted to the needs of the aging body. Nevertheless, specific lifestyle factors, such as obesity or physical inactivity, can lead to maladaptive immune aging, which can favor numerous chronic diseases [43].

#### 2.2.1. Aging of the Innate Immune System

The immune aging process of monocytes is partly reflected by alterations in the proportional distribution of monocyte subpopulations. The differential expression of CD14 and CD16 surface receptors are used to define major subsets of circulating cells. CD14++CD16- monocytes are often termed as classical monocytes, which typically represent up to 95% of the monocytes in a healthy individual. In contrast, CD16 expressing monocytes are divided into intermediate monocytes, which express CD14++CD16+, and non-classical subsets (CD14+CD16++) [44]. These monocytes differ in many aspects, including adhesion molecule and chemokine receptor (CCR) expression. The aging process is accompanied by an increase of non-classical CD14+CD16+ monocytes, which implicates an expansion of the more pro-inflammatory phenotypes associated with short telomeres. A successive shortening of the telomeres may lead to genomic instability, which favors cellular senescence or apoptosis [45].

Macrophages differentiate from circulating inflammatory or resident monocytes and are recruited to areas of tissue inflammation in response to pathogenic or tissue injury. Two major subtypes with an opposite inflammatory profile can be distinguished. While M1 macrophages represent a more pro-inflammatory subset which in turn produces pro-inflammatory cytokines, M2 macrophages act more immune-regulatory by producing anti-inflammatory cytokines and growth factors in order to promote wound healing and tissue remodeling. In aged individuals, an increased number of M1 macrophages can be found in different tissues, suggesting a contribution of these cells to a chronic pro-inflammatory state with aging [46]. The resulting dysbalance of the M1/M2 ratio is proposed to form the basis for the development of age-related diseases. For example, a significant association was found between macrophage polarization and proneness and atherosclerotic plaque development [47].

Neutrophils represent the most motile blood cells, and they possess the ability to move up a chemotactic gradient in an amoeboid fashion, termed chemotaxis, to the site of tissue infection or injury. It was reported that, during aging, these cells show a reduced chemotactic ability due to a reduced receptor expression and diminished intracellular signal transduction. This in turn leads to a diminished pathogen recognition, defective activation, and decreased phagocytosis [48]. Natural killer (NK) cells are innate lymphoid cells, which are classified into two distinct populations based on the surface density of their CD56 expression, namely CD56bright and CD56dim NK cells. CD56bright cells represent only about 10% of the circulating NK cells population and a major characteristic is there low cytotoxic response. Instead, these cells produce an array of cytokines and chemokines with mainly regulatory functions. In contrast, mature CD56dimCD16+ NK cells are predominant (90%) in the circulatory system and are potent mediators of natural and antibody-dependent cytotoxicity [49]. During aging, the absolute number of NK cells decreases as well as the NKbright:NKdim ratio, which is suggested to contribute to an impaired response to infectious diseases or cancers [50].

#### 2.2.2. Adaptive Immunity during Aging

The cells of the adaptive immune defense seem to be most affected by the aging process. While the number of T-cells remains largely constant, significant relative shifts in their subpopulations are recognizable. First, the ratio of the two major subsets, the CD4 T helper cells and the cytotoxic T-cells, is changing towards a reduced CD4/CD8 ratio. Within the CD4 and CD8 cells, both the proportion and absolute number of naïve T-cells decline stepwise with age. This is suggested to be the result of thymus involution and an insufficient homeostatic proliferation. While this reduction of cells is only modest for naïve CD4+ T-cells, it is striking for naïve CD8+ cells [51]. This is important since naïve T-cells represent a pool of antigen-inexperienced cells that ensure adequate immune responses against newly encountered pathogens. In contrast, effector T-cells accumulate in the elderly, especially terminally differentiated effector T-cells (T-EMRA), which have lost the expression of CD27 and CD28. The expansion of this cell type is mainly a result of senescence and cellular differentiation. Beside the effects of aging and lifestyle, the infection with cytomegalovirus (CMV) has been identified as an important driver of T-cells differentiation [52]. After the age of 65, the shift in ratio between naïve and differentiated T-cells affects both CD4 and CD8 cells, but age-related changes are more pronounced in CD8 than in CD4 cells [53]. This is particularly problematic from a health perspective because T-EMRA cells act vasoactively and may promote the development of arteriosclerosis [54], and additionally, their number correlates with insulin resistance in the presence of obesity [55].

### 2.3. The Aging Muscle and Sarcopenia

Normal aging leads to major physiological changes that can negatively affect physical function, including a progressive decrease in muscle mass, strength, and quality accompanied with an increase in fat mass. The aging process results in declines of muscle mass and strength by about 1% per year from the age of around 40 years referred to as sarcopenia, a clinical condition for which the implementation of an ICD-10-CM code as well as treatments have been developed in 2016 [56]. Muscle wasting, however, varies largely between individuals due to different changes related to the normal aging process per se but can be significantly modified by physical activity levels and exercise training [57]. For example, cross-sectional data showed that highly active older adults significantly delay accumulation of body fat and loss of lean mass in older age [58]. The data support the view that declines in physical function may not be related to age alone, but are rather confounded by muscle disuse and decreased levels of physical activity in the elderly general population [59,60].

According to population-based studies, the prevalence of sarcopenia in healthy adults aged 60 years and older is about 11% for men and 9% for women, respectively [61]. Major characteristics associated with aging are the muscle architecture and fiber type composition, tendon properties and vascular control in contracting muscle [62]. Age-related structural and functional changes in skeletal muscle contribute significantly to adverse health outcomes such as falls, fractures, functional impairments, and mobility limitations accompanied by elevated risk for hospitalization, morbidity, and mortality in older persons [63,64]. Sarcopenia is associated with anabolic resistance to feeding and exercise [65,66], changes in circulating hormones (mainly sexual hormones, growth hormone, and insulin-like growth factor 1, and insulin) [67], metabolic dysregulation and poor recovery from acute stress [68], low-grade systemic inflammation [69,70], mitochondrial dysfunction and cellular senescence [71], and impaired regeneration due to reduced stem cell function [72]. Recent results indicate that the immune system, especially macrophages, can have an important role in regulating muscle aging and contributes to sarcopenia through reductions in muscle stem cell populations [73]. Finally, increased amounts of adipose tissue often accompany sarcopenia, a condition referred to as sarcopenic obesity, associated with accelerated functional decline and poor health outcomes, compared to sarcopenia or obesity alone [74].

The decline in muscle mass, most prominent in the lower limbs, results from reductions in motor units and muscle fibers, with fewer fibers within a single motor unit [62]. While the size of type I muscle fibers is almost maintained, the size of type II fibers diminishes [75,76]. Nevertheless, the decline in muscle fiber number remains the main reason for the reduced muscle mass and strength with aging. Type II fibers seem to be particularly prone to increasing denervation with increasing age. This is further supported by a study that found that IIA and IIX myosin heavy chain mRNA and protein expression in human skeletal muscle decreased by 14% and 10%, respectively, per decade while type I mRNA remained constant with age [77]. However, fiber loss is at least partly prevented by the age-related remodeling of motor units that results in denervation of type II fibers with collateral re-innervation of type I fibers [78]. Whereas type I fibers are more susceptible to inactivity and denervation induced atrophy with a slow-to-fast fiber type shift, type II fibers are more affected with diseases (e.g., cancer, type 2 diabetes, chronic heart failure) and aging with a fast-to-slow fiber type shift [79].

Although the loss of motor neurons and muscle fibers and muscle fiber atrophy all contribute to sarcopenia, these are not the only factors explaining its occurrence. In fact, muscle strength decreases with age at a faster rate than muscle mass (i.e., 2.0–2.5% per year between the ages of 65 and 75) [80], representing a key characteristic of low muscle quality, defined as the ratio between muscle strength per unit of muscle mass, which has been found to be a better predictor of functional limitation and poor health in older adults than muscle mass alone [81,82]. Potential mechanisms include changes in muscle tissue composition and muscle cell metabolism based on high levels of inter- and intra-muscular adipose tissue and intramyocellular lipids [83]. Furthermore, the Baltimore Longitudinal Study of Aging found that body fatness is an early risk factor for accelerated decline in muscle quality [84]. On the other hand, recent findings indicate that older adults who maintain a high amount of physical activity have better mitochondrial capacity, similar to highly active younger adults, and this is related to their better muscle quality and physical performance [85]. This is of great importance since mitochondria are highly adaptive organelles and dynamically respond to environmental stimuli, such as nutritional status and physical activity, both of which play a key role in the development, prevention, and treatment of sarcopenia as well as of obesity [86,87,88].

Accordingly, measuring muscle strength, mass, and function in older persons helps clinicians to identify and caring older adults both at risk for or with sarcopenia and, more importantly, to facilitate early treatments, including exercise and nutritional interventions, that can help preserve muscle mass and function [89]. The European Working Group on Sarcopenia in Older People (EWGSOP) presented 2018 an update of the sarcopenia definition with low grip strength (<27 kg for men and <16 kg for women) as the first defining characteristic, with low muscle mass or quality proposed as a confirmatory finding and reduced physical performance as a sign of severe sarcopenia [90]. There are several tools that are used to assess muscle mass. No one technique meets all requirements, such as accuracy, feasibility, inexpensive, and ease of use, but dual energy X-ray absorptiometry (DXA) is often cited as the gold standard to measure muscle mass [91]. The use of magnetic resonance imaging (MRI) may provide a very good indication of the levels of fat/connective tissue infiltration into the muscle [92]. The measurement of muscle strength together with surrogate measures of muscle quality, such as phase angle [93] or ultrasound-measured thigh muscle echogenicity, could provide more information related to functional strength and may predict better clinical outcomes than either muscle mass or quality estimates alone [94]. Because of the interplay between multimorbidity and functional impairment [95], the implementation of easy-to-use physical performance tests that may reliably determine physical function of the patient has value in any phase of aging, but is particularly important in the elderly routine. A multicentre cohort study of 1098 participants aged ≥ 65 years in the Osteoporotic Fractures in Men Study recently found that walking speed was the strongest predictor of incident mobility disability in older, community-dwelling men [96]. Similarly, findings of a prospective study from 469,830 UK Biobank participants, aged 37 to 73 years from the general population, suggest that the combination of slow gait speed (≤0.8 m/s) and low muscle mass have the strongest associations with health outcomes and should be considered in the diagnosis of sarcopenia [97].

In time, large-scale clinical trials need to establish whether routine implementation of muscle strength and function development measures improves health outcomes in clinical and population settings. For the long term, it is important to incorporate physical performance assessment and promotion into healthcare in a manner that engages both clinicians and patients [89,98]. To understand how lifestyle factors can contribute to the decline in elderly skeletal muscle performance has major implications for health professionals who are developing therapeutic interventions aiming to improve muscle function and prevent physical disability with advancing age. The most important and highly accepted concepts relate to the lifestyle factors of diet and physical activity (see Section 5).

## 3. Physical Activity and the Gut Microbiota: Human Epidemiological Evidence

Different microbiota compositions were observed in subjects with different degrees of physical activity or fitness indicating that physical exercise may have a beneficial effect on the gut microbiota [99,100]. However, when groups with high and low activity or fitness level are compared, several other differences in lifestyle may influence the results, particularly diet, although more recent studies investigate independent effects of exercise. Additionally, in older adults, age-related changes, such as diseases, pain, movement restrictions, medication use, etc., can affect associations [101].

### 3.1. Association with α-Diversity Indices

Generally, a higher diversity of the gut microbiota was associated with a better health status [102]. Several studies in humans indicated a higher inter-individual variability and a lower diversity of the gut microbiota in older individuals [101], although the latter was not confirmed in all studies [103]. Studies in professional athletes showed that a high physical exercise level [104,105] is associated with a higher α-diversity of the gut microbiota compared to controls. Accordingly, the peak oxygen uptake, which is a measure of cardiorespiratory fitness and an indicator of physical fitness, was positively correlated with a higher α-diversity in healthy young adults [106]. In contrast, microbiota diversity did not differ between women with an active lifestyle compared to those with a more sedentary pattern [107].

In line with the finding in professional athletes [106], the peak oxygen uptake was positively correlated with α-diversity indices also in older adults who were included in a randomized cross-over trial but there was no difference in α-diversity between the exercise and the control period [108]. Additionally, other studies in older adults did not detect differences in the α-diversity by the physical activity level of the participants [109,110,111,112] (Table 1). It has been suggested that the biological age and not the chronological age is mainly associated with a lower gut microbiota diversity [10], which may be one reason why several studies failed to show an association of the physical activity level and the α-diversity of the microbiota in older adults. Besides the biological age, the weight status seems to influence the change of the α-diversity of the microbiota following exercise. In a large US study in older adults, the α-diversity was hardly different between individuals with different levels of exercise frequency but among the obese participants, those with a high exercise frequency had a higher α-diversity. Additionally, the α-diversity increased with increasing BMI [103], whereas in general, obesity is known to be associated with a lower diversity of the gut microbiota [113]. Thus, in older adults the weight status and the biological age of the individual seem to influence the association of the α-diversity and the exercise level among other factors. In this context, it should be noted that biological age has been shown to be increased by both long time obesity and sedentary lifestyles [114,115].

### 3.2. Association with the Gut Microbiota Composition

In general, differences in the composition of the gut microbiota in older compared to younger adults were reported. In particular, *Actinobacteria*, especially *Bifidobacterium*, and *Firmicutes,* seem to decrease, whereas *Bacteroidetes* and *Proteobacteria*, especially *Enterobacteriaceae* and *Clostridia*, increase in older age [101,116].

Several studies conducted in younger populations suggest that the exercise or fitness level can influence the gut microbiota composition [100,105,107,118,119]. In athletes, a lower abundance of *Bacteroidetes* and a higher abundance of *Firmicutes* were observed. Additionally, a higher proportion of *Akkermansia* was measured in athletes and in low BMI controls compared to high BMI controls [105]. This may be beneficial as previous studies indicated that *Akkermansia muciniphila* is inversely associated with obesity and metabolic disturbances [105], and a higher proportion was also observed in active compared to inactive women [107]. Consistent with the study in athletes [105], a trend of a lower abundance of the *Bacteroidetes* population was observed in active women who also showed a higher abundance of further potentially beneficial bacteria such as *Faecalibacterium prausznitzii*, *Roseburia hominis,* and *Bifidobacterium* spp. compared to inactive women [99,107]. However, it should be noted that in both studies [105,107], dietary intake strongly differed between active and sedentary groups.

In college students reporting high moderate-to-vigorous physical activity (MVPA) levels, *Paraprevotellaceae*, *Lachnospiraceae*, and *Lachnospira* were enriched, while among students reporting low MVPA, *Enterobacteriaceae* and *Enterobacteriales* were more prevalent [119]. In an intervention study with previously sedentary lean and obese young adults participating in endurance-based exercise training, exercise resulted in microbiota changes in *Collinsella* spp., *Faecalibacterium* spp., and *Lachnospira* spp. These changes were opposite depending on weight status [118], e.g., *Lachnospira* increased in lean participants as also shown in college students [119], whereas the prevalence was almost unaffected in participants with obesity [118].

In sedentary adults with type 2 diabetes or prediabetes, participating in an exercise training program increased the *Bacteroidetes* phylum and decreased the *Firmicutes*/*Bacteroidetes* ratio. Furthermore, a decrease in the *Clostridium* genus and *Blautia* was observed. Additionally, systemic and intestinal inflammatory markers were reduced, indicating a reduction in endotoxemia, which seems to be associated with a healthier microbiota [120].

Table 1 summarizes the results of studies conducted in older adults investigating differences of the gut microbiota composition by the physical activity or fitness level. Two of the studies reported changes following an exercise intervention [108,111]. In a cross-over trial including 62–76-year-old Japanese men, an endurance exercise program resulted in a decrease of *Clostridium difficile*, an enteropathogen, which is known for its toxin production. Additionally, the relative abundance of *Oscillospira* increased during exercise, although this was only detected in the control first group but not in the exercise first group. The observed changes in *Oscillospira* were associated with improvements of cardiometabolic markers. Although the effects were only modest, the results suggest beneficial effects of endurance exercise [108]. In the other intervention study, sedentary Japanese women aged 65 years and older received either aerobic exercise training, including brisk walking, or a trunk muscle training. *Bacteroides* increased only in the aerobic exercise group, in particular in those women who increased their time spent in brisk walking. Furthermore, increases in *Bacterioides* were associated with increases in cardiorespiratory fitness measured by a 6-min walk test [111]. The increase in *Bacteroides* is in line with a cross-sectional study in Finnish premenopausal women in whom a high cardiorespiratory fitness (high maximum oxygen intake) was associated with higher proportions of *Bacteroides* [121].

In the five observational studies, the relationship between the physical activity or fitness level and the gut microbiota composition was assessed. In Japanese community-dwelling older adults, only *Bacillaceae* and *Fusobacteriaceae* families were slightly different between the groups with low or high physical activity levels [109]. Two North-European studies compared community-dwelling older adults with high or low physical activity [110] or fitness levels [117]. Fart et al. [110] compared the gut microbiota between community-dwelling older adults and physically active senior orienteering athletes in Sweden. The higher activity level of the latter was associated with a potentially healthier microbiota as *Faecalibacterium prausnitzii,* which has been associated with favorable effects, was more prevalent, while *Parasutterella excrementihominis* and *Bilophila* unclassified, which may have negative effects on gastrointestinal health, were less prevalent [110]. In contrast, the Danish study in older adults assessed the fitness level based on a chair-rise test, BMI, and the DXA-measured leg-soft-tissue fat percentage, all of which were used to discriminate between groups. Results showed that the low and high fitness level groups also differed in the number of steps per day and duration of standing periods both of which were higher in the high fitness level group. The high fitness level was associated with a higher abundance of potentially beneficial bacteria such as of *Bifidobacterium adolescentis* and *Christensenella* species, whereas in the low fitness group pro-inflammatory *Enterobacterales* were enriched [117]. However, as the stratification criteria included anthropometric markers such as BMI, the impact of physical activity on the microbiota composition and on metabolic outcomes may have been overestimated [116]. A Chinese study included patients with hypertension who were divided into three groups according to their fitness level based on peak oxygen uptake levels [112]. In the group with the highest fitness level, the abundance of the class *betaproteobacteria*, the family *Ruminococcaceae* and the potentially beneficial genus *Faecalibacterium* were enriched compared to the other two groups. In patients with the lowest cardiorespiratory fitness, unfavorable microbiota members such as genus *Escherichia_Shigella* and the species *Escherichia coli* as well as the class *Bacilli*, the order *Lactobacillales*, the family *Lachnospiraceae*, the genera *Blautia* and *Ruminococcus*_sp__5_1_39BFAA showed a higher abundance. *Lactobacillales and Blautia* were positively associated with C-reactive protein [112]. In line with this latter observation, the abundance of *Blautia* was decreased by exercise training in sedentary adults [120]. In the observational US study of Zhu et al. [103], the microbiota structure of a large population consisting of 18–60-year-olds and older adults aged 61 years and over were examined. Among individuals of the older group, the gut microbiota was compared by exercise frequency, and the role of high or low exercise among overweight participants was considered. In individuals aged 61 and over, decreasing abundances of *Actinomycetaceae*, *Desulfovibrionaceae*, S24-7, *Pseudomonadaceae*, *Barnesiellaceae*, and *Oxalobacteraceae* and increasing abundances of *Campylobacteraceae*, *Fusobacteriaceae*, *Turicibacteraceae*, *Paraprevotellaceae*, *Clostridiaceae*, *Peptostreptococcaceae*, *Corynebacteriaceae*, and *Bacteroidaceae* were observed with increasing exercise frequency, respectively. Interestingly, a high exercise frequency seemed to shift the microbial composition closer to that of the younger adults aged 18–60 years. For example, the abundance of *Actinobacter* increased and that of *Cyanobacteria* decreased with increasing exercise frequency, both approaching that of younger adults aged 18–60. This is potentially beneficial because *Cyanobacteria* were associated with diseases. Moreover, it seems that regular exercise partially restores the abundances nearing that of normal weight elderly, and the authors conclude that regular exercise may decrease harmful and increase beneficial microbes in overweight elderly [103]. In line with this, a longitudinal cohort study in children and adolescents indicated that a high exercise level may have a protective role in terms of maintaining a healthy microbiota composition even if the children consumed a less healthy diet with low food diversity [100].

Overall, conflicting results relating to changes of gut microbiota composition induced by exercise in older adults may be due to weight status, metabolic and inflammatory state, as well as present diseases. Thus, more high-quality intervention studies are necessary to understand independent effects of physical activity on the gut microbiota composition in this heterogenous age group.

## 4. Physical Activity Shape the Immune System during Aging

In the ageing process, the immune system is restructured in many areas, which on the one hand represents an adaptation process, but on the other hand also entails a loss of function. Some significant lifestyle factors promote maldaptive changes in the immune system, accelerating some features of immune aging, promoting dysfunction and contributing to higher morbidity and mortality [122]. Conversely, the same lifestyle factors, such as physical activity, can positively influence the immune aging process. The following section addresses the impact of an active lifestyle on the aging immune system by outlining and discussing physiological interactions in light of the available literature [123,124,125].

### 4.1. Effects of Physical Activity on Aging Immune System

With regard to the cells of the innate immune system, several studies compared inactive elderly with physically active participants in cross sectional designs. Regular exercise in old age appears to be associated with enhanced NK cell and neutrophil functions, such as an increased cytotoxity and better migration of neutrophils toward IL-8 [126,127]. Furthermore, controlled intervention programs have indicated that exercise affects characteristics of innate immunity. For example, after ten weeks of high-intensity interval training, improvements in bacterial phagocytosis and oxidative burst of neutrophils have been observed [128]. Similarly, it was shown that the proportion of CD14+/CD16+ monocytes was reduced after twelve weeks of combined moderate strength and endurance training [129]. These findings imply a reduction in the pro-inflammatory and senescent subtypes of monocytes. Moreover, it is well known that regular exercise training leads to a reduction in visceral fat mass, ultimately reducing the infiltration of inflammatory monocytes into adipose tissue [130]. In addition, it has also been shown that physical exercise directly affects the conversion of M1 into M2 macrophages, which implicates a shift of the inflammatory milieu to lower secretion of pro-inflammatory cytokines [131]. Overall, these results suggest that an increase in habitual physical activity positively regulates factors of innate immune function, which could have a clinically effect with a reduced infection risk and a systemically lower inflammatory potential.

The majority of studies in the field exercise and immune aging focused on T-cells subpopulations. From a functional perspective, cross-sectional data demonstrated that older trained individuals showed an enhanced T-cell proliferation compared to untrained controls [126,132]. Spielmann et al. expanded this finding by demonstrating an association between fitness level, age and senescent T- cells. In detail, participants with above-average values of peak oxygen uptake (VO_2max_) showed 57% and 37% less senescent CD4+ and CD8+ T-cells, respectively, and 17% more naïve CD8+ T-cells [133]. Similar results were found for healthy older adults, maintaining high levels of aerobic fitness during the natural course of aging. Master athletes exhibit reduced hallmarks of immunosenescence, such as reduced senescent central-memory (CM), effector-memory, and highly-differentiated effector-memory T-cell phenotypes [134,135]. While some cross-sectional findings could not be proved by controlled exercise intervention studies [136,137], Philippe et al. demonstrated a proportional increase in naïve and CM T-cells after three weeks of endurance training in elderly prediabetic participants together with a decrease in senescent CD8+ EMRA T-cells [124]. Thus, endurance training and aerobic capacity in particular, contrary to resistance training, may have a powerful impact on the changes in T-cells and their subpopulations with aging [133,138,139]. Future longitudinal studies are needed to evaluate what kind of physical activity and exercise at which dose are most suitable for maintaining immune function during aging.

### 4.2. Immune Restorative Mechanisms of Physical Exercise

Potential mechanisms that mediate the immune regulatory effects of physical activity appear to arise out of the skeletal muscle itself. In particular, muscle contraction and an increased muscular energy metabolism lead to the production of several cytokines—so called myokines—or peptides with inflammatory regulatory potential [140]. One of the first and most effective myokine, which has been identified because of its immune regulatory function, is IL-6. The systemic secretion of IL-6 has a hormone-like effect in muscles and in other tissues and stimulates the production of immune-regulatory mediators, such as the IL-1 receptor antagonist and IL-10 [141]. Additionally, the downregulation of TNF-alpha was demonstrated by physical exercise and IL-6 infusion [142]. Besides IL-6, several other exercise-induced myokines may affect the aging immune system. For example, the hormone meteorin-like has been shown to induce adipose tissue browning, increase IL-4 levels, and promote the polarization of M2 macrophages [143]. IL-7 [144] and IL-15 [145] are myokines, which can stimulate lymphocyte proliferation. Therefore, it has been suggested that IL-7 exerts a protective effect on the thymus. Both IL-7 and IL-15 were found to be elevated in elderly participants with high levels of physical activity over lifespan [134]. IL-15 is further associated with an increased survival of naïve and memory phenotype CD8 + T-cells [146] and seems to have additional effects on immune homeostasis. Moreover, IL-15 reduces visceral and white adipose tissue accumulation by reducing pre-adipocytes [147].

A critical mechanism that can rejuvenate the aging immune system through regular physical activity appears to be adrenergic signalling. While it is not yet known whether myokines are released by exercise due to adrenergic signalling, various leukocyte subpopulations express high levels of adrenergic receptors, and the mobilization of these cells into blood during acute exercise seem to be affected by these signals [148]. However, the potential role of catecholamine signalling on long-term alterations of the leukocytes has not been extensively investigated. NK cells are the most responsive group of lymphocytes. Even a brief physical activity results in a four- to fivefold increase in the number of NK cells in peripheral blood [149]. The changes in NK cell activity in response to physical activity are mediated by the β2-adrenergic receptor (β-AR) subtype [150]. Mobilization of cytotoxic lymphocyte subtypes by catecholamines after acute bouts of dynamic exercise represent a possible mechanism for the protective effect of physical activity against age-related diseases such as cancer [151].

### 4.3. Clinical Effects of Exercise Training on Immune Function

With regard to upper respiratory tract infections, it has been convincingly shown that an active lifestyle can significantly reduce both the duration and severity of infections. Another indirect positive clinical effect of physical activity on the aging immune system is evident in the antibody response to vaccination [152,153]. For example, older people who performed moderate or intensive regular exercise showed a stronger antibody response after influenza vaccination than an inactive control group. Moderate activity performed 3 times per week over a 10-month period also increased vaccination response in elderly subjects measured as antibody titres [154]. This was also shown for a group of regularly active older female subjects who had better vaccination protection at 18 months after influenza vaccination compared to an inactive control group [125]. Epidemiological studies provide strong evidence that an active lifestyle has preventive and therapeutic effects on the development of tumours, especially breast and colorectal cancer. Data from animal models and “in vitro”provide further information that, at least partly, immunological processes might contribute, especially the increased function of NK cells as a result of sport. However, this topic is part of current research and will provide many more facts in the coming years [50,138].

## 5. Lifestyle Factors Affecting Elderly Skeletal Muscle Performance

### 5.1. Exercise and Muscle Strengthening

Low levels of physical exercise together with an unhealthy diet are major risk factors for sarcopenia, which together with other biological factors (e.g., hormones, inflammation and insulin resistance) and psychosocial factors (e.g., depression, social isolation, and loneliness) contribute to the decline in skeletal muscle performance with advancing age [155]. A wealth of evidence highlights the positive benefits of physical exercise in enhancing muscle function and/or preventing mobility and physical limitations [156,157,158]. The Lifestyle Interventions and Independence for Elders (LIFE) study showed that walking and low-intensity resistance training reduced the risk of major mobility disability in mobility-limited older adults over the course of two years in a dose-dependent manner when compared with a health education program with the greatest benefit by adding at least 48 min of physical activity to their weekly routine [159]. Though all types of physical activity offer benefits, resistance training is presently the most effective intervention to elicit improvements in muscle mass, strength, and function in older adults [160,161,162]. Indeed, a substantial part of the older population does benefit from a resistance-type exercise intervention, with more positive responses on lean body mass, muscle fiber size, muscle strength, and/or physical function following more prolonged exercise training [163]. A very recent meta-analysis on primary care interventions to address physical frailty among community-dwelling adults aged 60 years or older concluded that interventions using predominantly resistance-based exercise alone or with nutrition supplementation seem effective in reducing physical frailty and improving physical performance measured by gait speed, leg strength, the Short Physical Performance Battery (SPPB), and the Timed-Up and Go test, among other tests [164]. As an attempt to improve physical and functional capacity of older adults diagnosed with sarcopenia, the Vivifrail multicomponent tailored exercise program was developed (three times a week for 12 weeks, involving lower-limb muscles, upper body, and balance and gait retraining) [165]. Recent findings suggest that this type of intervention is highly effective in improving functional capacity as well as reducing falls risk in elderly ambulatory women with dynapenia thereby reducing the risk of frailty [166]. Although traditional slow-velocity resistance training is primarily associated with enhancements in muscle strength, also muscle power training with higher-velocity and lower-intensity (30–60% of one repetition maximum or the use of own body as resistance) is recommended to improve functional abilities (i.e., sit-to-stand, walking ability, stairs climbing) in elderly populations [167], due to its targeting of type II myofibers, which are more prone to atrophy in older adults (see Section 2.3).

Clinical studies have shown that the combination of resistance exercise and dietary protein supplementation is an effective strategy to prevent sarcopenia and improve physical functioning of older adults [168,169]. The aim of the ProMuscle in Practice Study was to test the effectiveness of a combined resistance exercise and dietary protein support intervention for community-dwelling older adults in a real-life setting [170]. After 12 and 24 weeks, the intervention, guided by physiotherapists and dietitians in the practice setting, resulted in a positive change in muscle mass, strength, and physical function and was found to have an 82.4% probability of being cost-effective [171]. So far, only few studies have been conducted to determine whether synergistic effects of exercise and nutritional interventions can result in sustained increases in physical performance beyond those produced by exercise alone. The Vitality, Independence, and Vigor in the Elderly 2 (VIVE2) Study was designed to examine the long-term effect of nutritional supplementation (150 kcal, 20 g whey protein, 800 IU vitamin D) plus structured exercise (three times a week for 24 weeks, involving walking and muscle strengthening activities) on physical function (gait speed, grip strength, SPPB) in mobility-limited older adults [172]. Findings from this trial and the large meta-analysis of Liao et al. [168] suggest that the training component per se is of primary importance when it comes to improving physical performance with no further improvement with added nutritional supplementation.

At the cellular level, mitochondrial function is closely linked to lifestyle. Impaired mitochondrial function plays an important role in the development and progression of sarcopenia and loss of physical function [173]. Both endurance and resistance training lead to an increase in mitochondrial quality [57,174], and this improvement in mitochondrial capacity is linked with better muscle quality, exercise efficiency, and physical performance in older adults [85,175]. Adaptations at the mitochondrial level may also explain important beneficial effects of resistance training [176,177]. As such, screening for mitochondrial function and targeting mitochondria with exercise may provide an effective way to evaluate individual responses to lifestyle interventions, thus contributing to a more tailored approach for mitigating sarcopenia and age-related declines in muscle function and performance.

In summary, multicomponent exercise and especially resistance training is a strong intervention for preserving functional capacity in older individuals. Thus, motivating older adults to be active can help to alleviate the loss in physical function associated with aging and, as such, support healthy aging. The new WHO Guidelines on Physical Activity and Sedentary Behaviour strongly recommend multicomponent physical activity that emphasizes functional balance and resistance training at moderate or greater intensity, on 3 or more days a week, to enhance functional capacity and to prevent falls in older adults [178].

### 5.2. Protein and Other Nutrients

With aging, the stimulating effects of both amino acid feeding and exercise on muscle protein synthesis become blunted, which is now widely believed to be a key factor responsible for age-related muscle loss [179]. However, performing exercise in close temporal proximity to protein ingestion and increasing the amount of protein ingested per meal (≥30 g), which contain higher amounts of leucine (≥2.5 g), can—at least to some extent—overcome anabolic resistance [180,181]. Evidence suggests benefits of dietary protein intake on preservation of muscle mass and bone health, thus attenuating risk of sarcopenia and bone loss [182,183], which is an important component of maintaining functional capacity in older individuals. On the basis of the results of a recent umbrella review, a significant effect of leucine supplementation on muscle mass is shown in persons with sarcopenia, but not in healthy subjects, whereas no clear effect of nutritional supplementation (e.g., protein, essential amino acid, leucine) has been reported on muscle strength and physical performance [184].

So far, only few studies have related protein intake to physical function and performance in aging. The Framingham Heart Study Offspring recently demonstrated in middle-aged and older adults that a higher-protein diet (≥1.2 vs. <0.8 g/kg body weight/day) across adulthood was associated with maintenance of physical function, which was based on self-reported measures, and lower odds of falls, fractures, and frailty over the span of two decades, especially in women [185]. The greatest risk reductions were found among those with higher protein intakes combined with either higher physical activity, more skeletal muscle mass, or lower body mass index [186]. Other longitudinal studies in community-based populations have used objective measures to assess function and/or performance and protein intake. The Quebec Longitudinal Study on Nutrition and Successful Aging (NuAge Study) was designed to examine the relation between mealtime distribution of protein intake and physical performance as assessed by a composite score of muscle strength (handgrip, arm, and leg) and mobility (chair stand, timed-up-and-go, and walking speed) and its 3-year decline in community-dwelling older adults [187]. Results showed that older men and women, with more-evenly mealtime distributed protein intakes (∼21, 29, and 30 g/meal and 18, 23, and 23 g/meal in men and women, respectively) had a higher muscle strength composite score, but not mobility score. However, more evenly distributed protein intakes were not associated with declines in strength and mobility throughout follow-up. Thus, findings suggest, first, that the consumption of a fair amount of protein at every meal, even if not reaching the proposed 30 g, is probably better for muscle health in older adults than ingesting only one daily high-protein meal. Second, the authors recommend that older persons should build up as much muscle mass and strength (functional reserve) early enough through an exercise programme in order to provide a buffer against age-related sarcopenia, thereby limiting the risk of major mobility disability. The proposed beneficial effects of a spread protein intake pattern over the main meals are in line with a recent cross-sectional study aimed to investigate whether protein intake and distribution are associated with muscle strength, physical function and quality of life in community-dwelling elderly people with a wide range of physical activity [188]. In the latter study, a more spread protein intake during the main meals was related to a higher gait speed, whereas a higher total protein intake was not associated with improved physical outcome measures. However, combining higher physical activity with higher total protein intake was related to a better quality of life, supporting the notion of a higher total protein intake together with an active lifestyle in the elderly.

Recently, a number of expert groups have advocated for higher daily protein intakes of 1.0–1.2 g/kg body weight/day in older adults, particularly to support the preservation of muscle mass and function [189,190]. Furthermore, data suggest that dietary leucine requirements of older men and women are almost twice as high as current recommendations, namely 78.5 mg/kg body weight/day [191], which is of particular interest as it is a powerful signal for stimulation of muscle protein synthesis in older adults [192,193]. Experimental evidence has shown that feeding elderly men protein at the RDA (0.8 g/kg body weight/day) for 10 weeks resulted in declines in whole-body and appendicular lean mass, while a diet providing twice the RDA recommendation for protein compared with the current guidelines was found to have beneficial effects on lean body mass and leg power in elderly men [194]. However, neither of the tests used to assess physical function (i.e., SPPB and timed-up-and-go) was altered by dietary intervention. On the other hand, recent findings of a randomized clinical trial of 92 men with physical dysfunction found that protein intakes equal to the RDA was sufficient to maintain lean body mass, muscle performance, and physical function over six months [195]. However, limited physical activity may be one of the reasons for the absence of treatment effects [196]. For example, the results of the study by Beelen et al. indicate that a higher amount of protein (1.5 g/kg body weight/day) without exercising may be not an effective approach to enhancing physical performance of older adults with already adequate protein intakes but a lack of physical activity [197]. In fact, it seems that just increasing protein without appropriate contractile manipulation might not be sufficient, possibly because low muscle mass may play a significant role [198]. Moreover, it can be assumed that individuals who are already at a nutrient level for optimum functioning may not benefit from an increase in protein intake [199].

All in all, it can be concluded that the effect of protein supplementation on skeletal muscle performance is minimal in older adults with already adequate protein intakes and limited physical activity. Nutritional intervention is probably an effective approach for malnourished patients [200], but most efficient when combined with a long-term exercise (resistance training) program, particularly in obese persons [184]. This simple measure helps to preserve muscular fitness in the elderly and, as such, promote healthy, active aging.

Nutrients that have been most consistently linked to the components of sarcopenia and muscle performance include apart from protein, long-chain polyunsaturated fatty acids, vitamin D, and multi-nutrients. A systematic review of 37 randomized, controlled trials that summarized the effect of combined exercise and various nutritional interventions including proteins, essential amino acids, creatine, β-hydroxy-β-methylbutyrate, vitamin D, multi-nutrients, or other in subjects aged 65 years and older noted that exercise beneficially affects muscle mass, strength, and physical performance, but the additive effect of nutritional interventions may be limited [201]. A novel randomized, double-blind, placebo-controlled supplementation trial in sarcopenic older individuals participating in a 12-week exercise training program found a significant beneficial effect of daily supplementation with whey protein (22 g), essential amino acids (including 4 g leucine), and vitamin D (100 IU) compared to placebo, with a gain of 1.7 kg in fat free mass, together with improvements in muscle strength, physical function, quality of life and inflammation [202]. Although the authors were not able to assess the effects of vitamin D supplementation separately from essential amino acid supplementation, this study suggests that whey protein, essential amino acid and vitamin D supplementation, together with resistance training, can improve skeletal muscle mass and performance in sarcopenic elderly.

It is now clear that vitamin D has important roles beyond its well-known effects on calcium and bone homeostasis. Several epidemiological studies have illustrated the potential role of vitamin D in order to maintain a good physical function in advanced age [203,204,205]. Vitamin D insufficiency (serum 25 (OH)D < 50 nmol/L) and deficiency (<25–30 nmol/L) are common in older people as a consequence of low dietary intake and reduced sunlight exposure, suggesting that vitamin D supplementation might represent an additional way, besides exercise training, to prevent sarcopenia and physical limitations, but the results are still inconclusive [206]. Meta-analyses of older adults have reported no significant effect of vitamin D supplementation on muscle strength and mobility [207]. However, results on muscle strength were significantly more important with people who presented a 25-hydroxyvitamin D level < 25–30 nmol/L [208,209]. Caution is warranted when administered at high dosages (e.g., 60,000 IU/month), which can even negatively affect muscle function and increase fall risk [210]. On the other hand, when resistance training is combined with vitamin D supplementation, in deficient individuals, the improvements in muscle strength and physical function are greater than exercise alone [211]. Therefore, older individuals should avoid vitamin D insufficiency and build up or maintain muscle mass and strength through resistance training in combination with adequate amounts of protein (>1 g/kg body weight/day) [212]. In fact, the PROVIDE Study showed that a vitamin D (800 IU/day) and leucine-enriched (3 g) whey protein (20 g) nutritional supplement consumed twice daily could improve measures of sarcopenia over a three-month intervention [213]. However, for individuals who are already at a nutrient level for optimum functioning (>50 nmol/L), further vitamin D supplementation would provide no additional benefit to muscle health [214]. Combining a multi-nutrient supplement containing proteins, creatine, vitamin D, and omega-3 fatty acids with home-based resistance training may offer a promising strategy to improve lean mass, muscle strength, physical performance, and muscle quality in free-living elderly, especially relevant for sarcopenic individuals with low physical activity levels [215].

Omega-3 polyunsaturated fatty acids (n3-PUFA) might be an alternative therapeutic agent for sarcopenia due to their anti-inflammatory properties, protein kinase activity (e.g., mTORC1) activation and reduction of insulin resistance [216]. N3-PUFA supplementation (dosages from 2.0 to 3.3 g/day over a 3–6-month time period) can attenuate the decline in muscle mass and function in healthy older adults [217]. In general, changes in muscle mass and function induced by n3-PUFA therapy are less than reported with resistance training alone [218]. However, in combination with an exercise intervention, n3-PUFA supplementation might augment the increase in muscle strength and function obtained by the exercise intervention [219]. Furthermore, a lifestyle intervention, including adequate fish intake (>500 g/week) and twice-weekly resistance training has been recently shown to trigger local anti-inflammatory and growth responses, thereby favoring skeletal muscle hypertrophy in older women [220], whereas older men may be less responsive to the anabolic sensitizing effect of n3-PUFAs [218,221].

In summary, it is essential to maintain good muscular fitness through an individually tailored exercise programme over the long-term in order to preserve functional capacity and muscle performance. Resistance training can build muscle mass and increase strength as we age, and therefore offers the most effective non-pharmacological intervention to prevent physical limitations with advancing age. Furthermore, increasing protein intakes in favour of plant-derived proteins [222] as well as a higher demand of specific nutrients (such as leucine, n3-PUFAs, and vitamin D) should be considered in older adults, especially in women, to overcome anabolic resistance and to further support exercise-induced adaptations and successful immune aging [2] (see Section 4).

### 5.3. Obesity and Weight Loss

Both obesity and malnutrition aare frequently observed in old age and are important determinants of functional impairments and frailty [223]. Obesity (excess fat mass), especially together with low muscle mass (called sarcopenic obesity), has been associated with several negative health outcomes, such as functional decline [224], increased risk of falls [225], osteoarthritis [226], and muscle weakness [227] accompanied by elevated risk for cardio-metabolic diseases and physical disability in older persons [228]. On the other hand, malnutrition is common among older people and often poorly recognized and underdiagnosed [229]. Insufficient dietary intake is not only related to the development of sarcopenia [86], but is also a major risk factor for cognitive or functional impairments and mortality in older patients [230,231].

Lifestyle interventions to combat obesity in older persons mainly focus on dietary approaches (e.g., caloric restriction, high-protein diet) and exercise consisting of aerobic endurance training and resistance training [232]. Although a caloric restriction weight loss diet is the first therapeutic option for the treatment of obesity, it is associated with loss of muscle and bone mass [233], which can further worsen sarcopenia and increase fracture risk [234]. Furthermore, there is limited evidence that caloric restriction alone demonstrates improvements in measures of physical performance. In fact, weight loss plus combined aerobic and resistance exercise was the most effective strategy in improving physical fitness of obese older adults during weight loss [162]. Moreover, adequate protein intake (∼1.2 g/kg body weight/day) combined with vitamin D and calcium supplementation is recommended in obese older adults undertaking caloric restriction and resistance training in order to preserve muscle mass and bone mineral density during weight loss [235]. Finally, resting metabolic rate (RMR) is lower in older adults, which is mostly attributable to changes in body composition [236], and may predispose to future weight gain. However, regular exercise training has the potential not only to maintain muscle mass and RMR with aging [237], but also to lower the risk of inadequate dietary intake that may influence the onset and the course of physical limitations [238].

## 6. Nutritional Considerations to Support Healthy Active Aging: Focus on Gut Microbiota

Nutritional risk represents a fundamental factor influencing the aging trajectory [239]. Age itself and several age-related conditions, including frailty, chronic illnesses, cognitive decline and depression, are in fact associated with malnutrition [239]. Thus, any effective strategy for promoting healthy active aging should carefully consider tailored interventions against malnutrition [240,241]. The intestinal microbiota from malnourished individuals is deeply disrupted and contributes to wasting through multiple mechanisms, including anabolic resistance, malabsorption, induction of anorexia and reduced synthesis of vitamins [242,243,244,245]. Any nutritional strategy preventing malnutrition and supporting healthy active aging should thus look at the effects on microbiota carefully.

Moderately-high protein intake in the range of 1–1.2 g/kg body weight/day is generally considered one of the cornerstone nutritional measures in older individuals, for its capacity of promoting protein synthesis and preventing physical frailty [241]. However, from a microbiota perspective, data from animal models indicate that increasing protein intake, especially of animal origin, is associated with induction of dysbiosis, depletion of bacterial taxa producing SCFA and increased production of trimethylamine N-oxide (TMAO), a marker of increased cardiovascular risk [246,247,248,249]. Similar findings were also obtained in human athletes fed with high-protein diets [250,251] and in patients with non-alcoholic fatty liver disease [252]. Thus, the possible benefits on metabolic and muscle health by high-protein diets could be undermined by stimulation of inflammation and increased cardiovascular risk, at least in experimental conditions.

Other studies have, however, elucidated that, in practice, the gut microbial communities are not just shaped by dietary protein quantity, but are also influenced by their quality [253], timing of consumption [254], availability of nitrogen, and level of intake of other nutrients, including carbohydrates and fibers [255,256]. Accordingly, two different randomized controlled trials conducted in older persons and one intervention study in sedentary adults have shown that, in real-life conditions, increasing protein intake is not associated with detrimental changes of gut microbiota composition, apart from a small decrease in bacterial taxa producing SCFA [257,258,259]. The marker of cardiovascular risk TMAO is significantly increased only in older subjects with extremely high protein intake levels (>1.6 g/kg body weight/day), which should be avoided [260,261]. Thus, in a healthy aging perspective, protein intake should not exceed 1–1.2 g/kg body weight/day, be sustained particularly by proteins with high biological value, such as casein and whey proteins, and be associated with an adequate intake of carbohydrates and fibers [262].

Fiber intake is in fact able to promote gut microbial diversity and representation of Bifidobacteria and species producing SCFA [263], as demonstrated in different population-based studies and experimental models [264]. SCFA, and particularly butyrate, once absorbed into systemic circulation, exert a wide range of metabolic functions including anabolic regulation, insulin sensitivity and modulation of inflammation [265]. Anabolic resistance, oxidative stress and inflammaging are deeply involved in defining the aging trajectory, and an adequate synthesis of SCFAs by the intestinal microbiota could contribute to mitigate these pathways [266]. Fruit and vegetables, the main dietary sources of fibers, also contain high amounts of polyphenols, which are metabolized by gut microbial species. A polyphenol-rich dietary pattern can also contribute to modulate oxidative stress and counteract age-related gut microbiota dysbiosis [267], but most importantly, improve intestinal permeability [268], which is strictly associated with chronic activation of the inflammatory response [266,269]. Thus, an elevated intake of fruit and vegetables should be part of any anti-aging dietary recommendation.

Overall, the principles of a diet with a balanced content of proteins with high biological value and high intake of complex carbohydrates, fibers and polyphenols are embodied in the Mediterranean diet. Large studies have demonstrated that high level of adherence to Mediterranean diet is associated with gut microbiota diversity, improved balance between symbionts and pathobionts, and higher microbiome production of SCFA [270,271,272]. Namely, the NU-AGE Study demonstrated significant microbiome-based benefits of adopting a Mediterranean dietary pattern in older fit or pre-frail subjects, improving inflammation, frailty, and cognitive performance at one-year follow-up [272]. SCFA represent central mediators of the health benefits of Mediterranean diet, since their production by gut microbial communities is increased after only few months of dietary intervention, even before major changes in microbiota composition can be detected [273]. The anti-aging activity of SCFAs in Mediterranean diet is not limited to modulation of inflammation and anabolism, but also to cognitive function, since they are associated with reduced amyloid deposition in dementia [274]. The anti-aging activity of the Mediterranean diet is, however, also sustained by other nutrients, including polyphenols, polyunsaturated fatty acids, vitamins, and minerals, and acts globally towards the reduction of inflammation and oxidative stress, and the improvement of immune function, genomic stability, and insulin sensitivity [275,276,277]. Thus, from a microbiome perspective, the Mediterranean diet should be considered, according to the current state of knowledge, the best anti-aging dietary pattern.

## 7. Conclusions and Future Perspectives

The concept of a healthy resilient gut microbiome relies on its high richness and biodiversity. The intestinal microbiota plays an important role in many metabolic processes that are beneficial to the host such as synthesis of vitamins and production of SCFA. On the other hand, it has also been associated with chronic illness and geriatric conditions, which can be positively influenced by lifestyle changes, thereby supporting healthy aging. The present review aimed to summarize the current literature on the role of physical activity and specific exercise and dietary interventions in the composition of the gut microbiota. The effects of other lifestyle factors, such as stress, drug intake, smoking habits, and sleep, and other environmental factors, including the geographical area and air pollutants that can affect the gut microbiota, have been reviewed recently [278]. A poor lifestyle characterized by an unbalanced diet and sedentarism, but also other factors, such as physiological or psychological stress, chronic intake of drugs, and a lack of sleep can lead to gut dysbiosis, promoting inflammation and the development of chronic diseases that can negatively influence muscle mass and function, particularly during aging [279]. On the other hand, a healthy lifestyle can play a significant role in reducing the hallmarks of immune aging. In particular, regular endurance exercise is an effective strategy for supporting successful immune aging and the most promising approach to counteract cellular immunosenescence and inflammaging. In comparison, resistance training is a strong intervention for preserving functional capacity and muscle performance in older individuals and is therefore the most effective non-pharmacological intervention to prevent physical limitations with advancing age. Adherence to a Mediterranean-style diet, with a high intake of proteins, fibers, and polyphenols, should be considered in older adults to further support exercise-induced adaptations and to prevent age-related non-communicable diseases. Health benefits could partly depend on the effect of such interventions that influence the biodiversity and functionality of intestinal microbiota, promoting the synthesis of metabolically active mediators such as butyrate and other SCFA. Although a high-protein intake is generally associated with microbiota dysbiosis in animal models, the benefits of increased muscle protein synthesis in older humans could outweigh the effects on microbiota, which need further investigation in the future. Furthermore, there is limited research available on how adaptations to exercise impact the gut microbiota in older individuals. Despite some studies that have shown aerobic exercise to beneficially alter gut microbiota composition, functional capacity, and metabolites, the effects of different exercise modalities, frequencies, and intensities remain unknown. Future research should differentiate between short-term and long-term effects. Another important question to be answered is: how does exercise interact with nutritional factors, such as restricted energy, higher protein consumption, or probiotics, in shaping the gut microbiome? Additionally, conflicting results exist relating to changes of gut microbiota composition between more and less active elderly, which may be due to differences in weight status, metabolic and inflammatory state and present diseases. In order to gain a better understanding of the precise lifestyle recommendations needed to maintain gut and immune health, the inclusion of several lifestyle determinants (e.g., background diet and body composition, level of physical activity, smoking habits, drug consumption, and place of living) should be included in future longitudinal studies. As such, more high-quality intervention studies are necessary to understand potential independent effects of lifestyle and diet in older adults in order to confirm and quantify the possible associations between microbiota and trajectories of aging.

## Figures and Tables

**Figure 1 nutrients-13-02045-f001:**
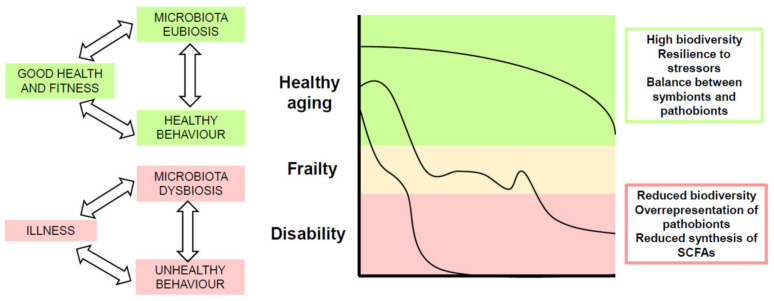
Hypothetical model of the possible association between microbiota and trajectories of aging, based on the current state of knowledge on the association between frailty and microbiota. Healthy aging (curve on the top of the graph) may be associated with maintenance of microbiota eubiosis (i.e., balance between symbionts and pathobionts) that contributes to the overall fitness of the organism in a virtuous cycle. Aging with frailty, characterized by a general, slow decline of health status and performance following acute illness and exacerbations of chronic diseases with occasional and transitory improvements (curve in the middle), may be associated with tendency towards microbiota dysbiosis (i.e., reduced species richness and increased representation of opportunistic pathogens). Disability, following an acute disruptive event or accelerated multimorbidity (curve on the bottom of the graph), may be associated with severe microbiota dysbiosis, leading to a vicious cycle that sustains illness and further decline of health status.

**Table 1 nutrients-13-02045-t001:** Intervention and Observational Studies (modified, based on [116]).

Reference	Subjects, Age (N), Country	Methodological Approach	Main Findings in Gut Microbiota Composition	Additional Observation/Outcome, if Applicable
Intervention Studies				
Taniguchi et al. 2018 [108]	62–76 years old men (33), Japan	Randomized crossover trial. 3 sessions/week exercise in a cycle-ergometer and control sedentary period (5 weeks each)	Exercise vs. control period: ↑ *Oscillospira* (in control period first only) ↓ *Closstridium difficile* No difference in α-diversity (Shannon, OTUs) between exercise and control period but baseline peak oxygen uptake was positively correlated with α-diversity indices.	Changes: correlated with cardiometabolic phenotypes. Changes in diversity indices: negatively associated with changes in SBP and DBP during exercise periods.
Morita et al. 2019 [111]	>65 years old women (32), Japan	Non-randomized comparative trial. 12 weeks of 1 h/daily brisk walking vs. 1 h/weekly session of trunk muscle training	Post- vs. Pre-Intervention: Brisk walking ↑ *Bacteroides* ↓ *Clostridium subcluster XIV* Trunk muscle training ↓ *Clostridium subcluster IX* *Bacteroides* before intervention and change in 6MWD were independent contributors of change in *Bacteroides*. Stronger increases of *Bacteroides* in subjects who increased the daily time spent in brisk walking for 20 min. or more vs. in those who did not.	Constipation assessment scale was slightly improved in brisk walking group; increased cardio-respiratory fitness in both groups
Observational Studies				
Zhu et al. 2020 [103]	Older adults, ≥61 years, (897; including 413 with overweight) USA	4 groups by exercise frequency: daily, regular, occasional, never/rare *	α-diversity (Shannon, OTU numbers) almost unaffected by exercise frequency if all older adults were considered	α-diversity (Shannon, OTU numbers) increased with increasing BMI
		Considering only overweight individuals (OE): classified by exercise frequency to daily or regular (DROE) vs. never or rare exercise (NROE)	DROE vs. NROE: ↑ α-diversity ↑ *Bacteroidetes*, *Cyanobacteria*, *Firmicutes*, *Tenericutes*, *Verrucomicrobia* ↑ *Turicibacteraceae* ↓ *Pseudomonadaceae*, *Oxalobacteraceae*, *Odoribacteraceae*, *Barnesiellaceae*	-
Aoyagi et al. 2019 [109]	65–92 years old(140 M and 198 F), Japan	Monitoring (1 month) of daily physical activity. More vs. less active groups (≥ vs. <15 min/d at >3 METS or ≥ vs. <7000 steps/d)	More active vs. less active group: ↑ *Bacillaceae* ↓ *Fusobacteriaceae*	Intestinal health measured as infrequent bowel movement/defecation frequency was better in the more active group
Fart et al. 2020 [110]	≥65 years old men and women (53 M and 45 F), Sweden	Physically active senior orienteering athletes vs. community-dwelling older adults	Active seniors vs. community-dwelling older adults: ↑ *Faecalibacterium prausnitzii* ↓ *Parasutterella excrementihominis* ↓*Bilophila* unclassified No difference in Shannon diversity index	-
Yu et al. 2018 [112]	65–80 years old patients with hyper-tension (32 M and 24 F), China	Patients classified according to Weber’ system for functional capacity. Normal exercise capacity vs. reduced exercise capacity	No significant difference in α-diversity measures (Chao 1, Simpson, Shannon) Subjects with reduced exercise capacity: ↓ *Betaproteobacteria*, *Burkholderiales, Alcaligenaceae* *Lactobacillales, Blautia, Rumino-coccus*_*sp._5_1_39BFAA* and *E. coli* were negat ively correlated with peak VO2/kg; *Alcaligenaceae* was positively correlated with peak VO2/kg levels	Increased CRP in reduced exercise capacity. Positive association with CRP: *Lactobacillales, Eubacterium_* *hallii*_group, *Blautia*. Negative association with CRP: *Alcaligenaceae*
Castro-Mejía et al. 2020 [117]	>65 years old men and women (109 M and 98 F), Denmark	Community-dwelling older adults with high high vs. low physical fitness	High vs. low physical fitness: ↑ *Bifidobacterium adolescentis* ↑ *Christensenella* species High-fitness subjects, despite higher energy intake, had: leaner bodies, lower fasting proinsulin-C-peptide/blood glucose levels likely driven by higher dietary fiber intake, physical activity and increased abundance of *Bifidobacteriales* and *Clostridiales* species and associated metabolites (i.e., enterolactone) explaining 50.1% of the individual variation in physical fitness.	Abundance corresponded negatively with proinsulin, HbA1c, VLDL, triglycerides

Changes (↑: increase; ↓: decrease) in the relative abundance of selected microbial taxa and in bacterial diversity with the interventions/more active groups; * never, rare (a few times/month), occasional (1–2 times/week), regular (3–5 times/week), and daily. Abbreviations: 6MWD, distance in 6-min walk test; CAVI, cardio-ankle vascular index as a marker of arterial stiffness; CRP, C-reactive protein; DBP, diastolic blood pressure; F, female; HbA1c, hemoglobin A1c; M, male; SBP, systolic blood pressure; VLDL, very low density lipoprotein; vs., versus.

## Data Availability

Data sharing is not applicable to this article.

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
