# Peer review of "The Effects of Lifestyle and Diet on Gut Microbiota Composition, Inflammation and Muscle Performance in Our Aging Society"

_nutrients, 2021, doi:10.3390/nu13062045_

Round 1

Reviewer 1 Report

The present manuscript reviews the effects of physical activity and specific exercise and dietary interventions on gut microbiota composition, inflammation and muscle performance in the aging population.

It is a comprehensive manuscript, and i agree with the authors statement that usually only one of the mentioned lifestyle factors are addressed in intervention studies, while combined interventions are very rarely applied. Besides, there is indeed a need to develop a broad understanding of the connection between microbiome, lifestyle and diet, and it is a crucial factor for maintaining good health in our aging society.

Literature is up to date and relevant.

The present manuscript as i mentioned in my previous report it is interesting, of high interest and pretty complex.   There is indeed a need to develop a broad understanding of the connection between microbiome, lifestyle and diet, and it is a crucial factor for maintaining good health in our aging society.The impact of environmental factors, including aspects of lifestyle, on the microbiota is particularly poorly understood, therefore would be recommanded maybe to the authors to discuss a bit more in details about this issue. A description of some areas that should be addressed in future research is also beneficial to be presented.   If these aspects will be further addressed in this manuscript than no further objections from my side.

Author Response

The authors would like to thank Reviewer 1 for evaluating our manuscript and providing expert opinions, which have helped us, improve our work.

The impact of environmental factors, including aspects of lifestyle, on the microbiota is particularly poorly understood, therefore would be recommended maybe to the authors to discuss a bit more in details about this issue. A description of some areas that should be addressed in future research is also beneficial to be presented.

A: Thank you for your suggestions. We added in the conclusion section some information about environmental factors that can affect the gut microbiota and some areas that should be addressed in future research.

The present review aimed to summarize the current literature on the role of physical activity and specific exercise and dietary interventions in the composition of the gut microbiota. The effects of other lifestyle factors, such as stress, drug intake, smoking habits and sleep, and other environmental factors, including the geographical area and air pollutants that can affect the gut microbiota, have been reviewed recently by Redondo-Useros et al [278]. A poor lifestyle characterized by an unbalanced diet and sedentarism, but also other factors, such as physiological or psychological stress, chronic intake of drugs, and a lack of sleep can lead to gut dysbiosis, promoting inflammation and the development of chronic diseases that can negatively influence muscle mass and function, particularly during aging [279]. On the other hand, a healthy lifestyle can play a significant role in reducing the hallmarks of immune aging.

Despite some studies have shown that aerobic exercise beneficially alters gut microbiota composition, functional capacity, and metabolites, the effects of different exercise modalities, frequencies, and intensities are unknown. Future research should differentiate between short-term and long-term effects. Another important question to be answered is: how does exercise interact with nutritional factors, such as restricted energy, higher protein consumption or probiotics in shaping the gut microbiome? In order to gain a better understanding of the precise lifestyle recommendations needed to maintain gut and immune health, the inclusion of several lifestyle determinants (e.g., background diet and body composition, level of physical activity, smoking habits, drug consumption, and place of living) should be included in future longitudinal studies.

Reviewer 2 Report

The Effects of Lifestyle and Diet on Gut Microbiota Composition, Inflammation and Muscle Performance in our Aging Society

Barbara Strasser, Maike Wolters, Christopher Weyh, Karsten Krüger and Andrea Ticinesi

This review covers many topics, separated into the following chapters:

  • Effects of aging on the microbiota, immune system (innate and adaptive) and skeletal muscle, homing in on sarcopenia.
  • Associations between activity and microbiota composition in older adults
  • Gains in immune protection and microbiota markers through physical activity in older adults
  • Effects of exercise and protein intake on skeletal muscle performance

They end the paper with recommendations for healthy aging with consideration for the microbiome.

This is a solid review, covering many aspects of the subject from many sources and viewpoints with inclusion of very recent research, not much new information on musculoskeletal side. They also cover contraindicatory studies, showing the discordance within the field and where clearer research is needed. It is a good overview, bringing together many research topics in one place. No objections to publication.

Minor Comments

  • Line 158 – mention of telomeres, no follow-up
  • Line 199 – exact title of paper cited, should be paraphrased a little, also not sure of context
  • Figure 1 – not sure which information this curve is based on

Citation 246 – spelling mistake

Author Response

The authors would like to thank Reviewer 2 for evaluating our manuscript and providing expert opinions, which have helped us, improve our work.

Line 158 – mention of telomeres, no follow-up

A: We added some information about the physiological meaning of shortened telomeres. A successive shortening of the telomeres may lead to genomic instability, which favors cellular senescence or apoptosis.

Line 199 – exact title of paper cited, should be paraphrased a little, also not sure of context

A: Thank you for this information. We changed it accordingly: Beside the effects of aging and lifestyle, the infection with cytomegalovirus (CMV) has been identified as an important driver of T- cells differentiation”.

Figure 1 – not sure which information this curve is based on

A: We revised the text referring to Figure 1 and the caption of the figure to improve comprehension by readers. The figure represents a hypothetical model, based on the results of the main studies exploring the relationship between frailty and microbiota. So, it is not based on original data.

Citation 246 – spelling mistake

A: Thank you. We corrected reference 246.